# FEAT2VEC: DENSE VECTOR REPRESENTATION OF DATA WITH ARBITRARY FEATURES

## ABSTRACT

Methods that calculate dense vector representations for features in unstructured data—such as words in a document—have proven to be very successful for knowledge representation. We study how to estimate dense representations when multiple feature types exist within a dataset for supervised learning where explicit labels are available, as well as for unsupervised learning where there are no labels. Feat2Vec calculates embeddings for data with multiple feature types enforcing that all different feature types exist in a common space. In the supervised case, we show that our method has advantages over recently proposed methods; such as enabling higher prediction accuracy, and providing a way to avoid the cold-start problem. In the unsupervised case, our experiments suggest that Feat2Vec significantly outperforms existing algorithms that do not leverage the structure of the data. We believe that we are the first to propose a method for learning unsupervised embeddings that leverage the structure of multiple feature types.

## 1 INTRODUCTION

Informally, in machine learning a *dense representation*, or *embedding* of a vector $\vec{x} \in \mathbb{R}^n$ is another vector $\vec{y} \in \mathbb{R}^r$ that has much lower dimensionality ($r \ll n$) than the original representation, and can be used to replace the original vector in downstream prediction tasks. Embeddings have multiple advantages, as they enable more efficient training (Mikolov et al., 2013), and unsupervised learning (Schnabel et al., 2015). For example, when applied to text, semantically similar words are mapped to nearby points.

We consider two kind of algorithms that use embeddings:

1. Unsupervised methods (sometimes referred as self-supervised methods) like Word2Vec (Mikolov et al., 2013), are designed to provide embeddings that are useful for a wide-array of predictions tasks. For example, the loss function of the continuous bag of words (CBOW) algorithm of Word2Vec is tuned to predict the next word of a sequence; however, in practice, the embeddings produced are mostly used for other tasks, such as analogy solving (Mikolov et al., 2013), or sentiment analysis (Le & Mikolov, 2014). In the context of this paper, we refer to the embeddings of an unsupervised method that can be used for a variety of auxiliary prediction tasks as *general-purpose*.

2. Supervised methods, like matrix factorization, produce embeddings that are highly tuned to a prediction task. These embeddings may be interpretable but do not usually generalize to other tasks. We refer to these embeddings as *task-specific*. Matrix factorization and Word2Vec are unable to calculate embeddings for items that are not available during training ("cold-start" problem). While recent work using n-gram features (Bojanowski et al., 2016) have addressed this limitation for supervised and unsupervised tasks, it can only be used for a single feature type—words.

In this paper we propose Feat2Vec as a novel method that allows calculating embeddings of arbitrary feature types from both supervised and unsupervised data. Our main contributions are:

- Unsupervised Feat2Vec. Existing general-purpose dense representation methods are largely restricted to one or two feature types. For example, the Word2Vec methods can only calculate embeddings for words, while follow-up work has enabled embeddings for

both words and documents (Le & Mikolov, 2014). To our knowledge, Feat2Vec is the first algorithm that is able to calculate general-purpose embeddings that are not tuned for a single specific prediction task for arbitrary feature types.

- Supervised Feat2Vec. Task-specific methods can use arbitrary feature types, but are restricted in that embeddings must be calculated for each individual feature, while sometimes higher-level of abstractions may be desirable—for example, we may want to have embeddings of documents instead of simply words. This capability makes Supervised Feat2Vec extremely flexible. We demonstrate that our method can be used to calculate embeddings of unseen (cold-start) items when there is an alternative textual description.

## 2 PRELIMINARIES

Factorization Machine (Rendle, 2010) is one of the most successful methods for general-purpose factorization. Rendle (2010) formulated it as an extension to polynomial regression. Consider a degree-2 polynomial (quadratic) regression, where we want to predict a target variable $y$ from a vector of inputs $\vec{x} \in \mathbb{R}^n$:

$$\hat{y}(\vec{x}; \vec{b}, \vec{w}) = \omega\big(b_0 + \sum_i b_i x_i + \sum_{i=1}^{n} \sum_{j=i+1}^{n} w_{i,j}\ x_i x_j\big) \tag{1}$$

In words, $n$ is the total number of features, the term $b_0$ is an intercept, $b_i$ is the strength of the $i$-th feature, and $w_{i,j}$ is the interaction coefficient between the $i$-th and $j$-th feature. The function $\omega$ is an activation. Choices for $\omega$ include a linear link ($\omega(x) = x$) for continuous outputs, or a logistic link ($\omega(x) = \frac{\exp(x)}{\exp(x)+1}$) for binary outputs.

Factorization Machine replaces the two-way individual pairwise parameters $w_{i,j}$ for each interaction with a vector of parameters $\vec{w}_i$ for each feature. This is a rank-$r$ vector of latent factors—*embeddings* in the neural literature—that encode the interaction between features and replaces the quadratic regression model with the following:

$$\hat{y}(\vec{x}; \vec{b}, \vec{w}) = \omega\big(b_0 + \sum_i b_i x_i + \sum_{i=1}^{n} \sum_{j=i+1}^{n} (x_i \vec{w}_i) \cdot (x_j \vec{w}_j)\big) \tag{2}$$

Intuitively, the dot product ($\cdot$) returns a scalar that measures the (dis)similarity between the latent factors of features $x_i$ and $x_j$. Polynomial regression has $n^2$ interaction parameters, and Factorization Machine has $n \times r$. While setting $r \ll n$ makes the model less expressive, factorization will typically exploit features having some shared latent structure. Factorization Machine may dramatically reduce the number of parameters to estimate. Rendle (2010) shows that when the feature vector **x** consists only of two categorical features in one-hot encoding, Factorization Machine is equivalent to the popular Matrix Factorization algorithm (Koren et al., 2009).

## 3 FEAT2VEC

We now describe how Feat2Vec extends the Factorization Machine model by allowing grouping of features, and enabling arbitrary feature extraction functions (§ 3.1). We also report a supervised method to learning Feat2Vec (§ 3.2), as well as a novel unsupervised training procedure (§ 3.3).

### 3.1 MODEL

We propose a framework for extending factorization machine with neural methods, by introducing structure into the feature interactions. Specifically, we do this by defining *feature groups*, $\vec{\kappa}$, where each group contains features of a particular type. Explicitly, $\vec{\kappa}$ is a partition of the set of feature columns in a dataset and each set within the partition is a feature group. The embeddings of a feature group are then learned via a *feature extraction function*, $\phi_i$, defined for each feature group. Feat2Vec will then extract features from each feature group, and build $r$ latent factors from them. In Factorization Machine, all the feature embeddings interact with each other, while in Feat2Vec, the interactions only occur between different feature *groups*.

Formally, the addition of deep extraction methods yields the following statistical model:

$$\hat{y}(\vec{x}, \vec{b}, \vec{\phi}) = \omega\left(b_0 + \sum_{i=1}^{n} b_i x_i + \sum_{i=1}^{|\vec{\kappa}|} \sum_{j=i}^{|\vec{\kappa}|} \phi_i(\vec{x}_{\vec{\kappa}_i}) \cdot \phi_j(\vec{x}_{\vec{\kappa}_j})\right) \tag{3}$$

In this notation, $\vec{x}_{\vec{\kappa}_i}$ is a subvector that contains all of the features that belong to the group $\vec{\kappa}_i$. Thus, $x_{\vec{\kappa}_i} = [x_j : j \in \vec{\kappa}_i]$. The intuition is that by grouping (sub-)features as a single entity, we can can reason on a higher level of abstraction. Instead of individual sub-features interacting among each other, the embeddings of feature groups interact with those of other groups. $\phi_i$ is a feature extraction that inputs the $i$-th feature group of the instance, and returns an $r$-dimensional embedding. The feature extraction function $\phi_i$ can allow for an arbitrary processing of its subfeatures. Across groups, entities interact with each other via the output of $\phi$ only.

As a concrete example of an application of this grouping/feature extraction, we might group the individual words of a document into a "document" feature group, and allow this document embedding to then interact with learned embeddings of other document metadata (such as author id). We might expect the extraction function $\phi$ for the words in a document to extract features that characterize the attributes of the document taken as a whole, rather than simply the sum of its individual words.

Figure 1 compares existing factorization methods with our novel model. In this example, Feat2Vec is using two feature groups: the first group only has a single feature which is projected to an embedding (just like a regular Factorization Machine); the second group has multiple features, which are together projected to a single embedding.

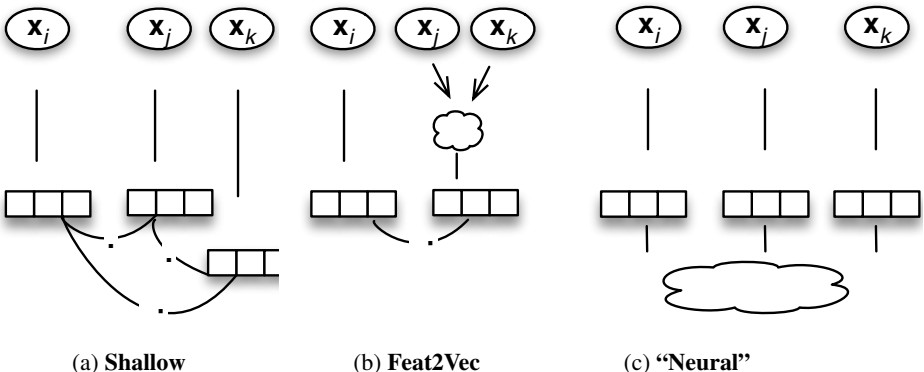

(a) **Shallow**    (b) **Feat2Vec**    (c) **"Neural"**

Figure 1: **Network architectures for factorization models**. The white clouds ($\circlearrowleft$) represent deep layers, for example a convolutional network for text features.

The simplest implementation for $\phi_i$ is a linear fully-connected layer, where the output of the $r$-th entry is:

$$\phi_i(\vec{x}_i; \vec{w})_r = \sum_{a=1}^{d_i} w_{r_a} x_{i_a} \tag{4}$$

Note that without loss of generality, we could define a model that is equivalent to a shallow Factorization Machine by allowing each feature group to be a singleton : $\vec{\kappa} = \{\{x_1\}, \{x_2\} \dots \{x_n\}\}$ and the linear extraction function presented in Equation 4.

We can use Feat2Vec to both use large feature sets and overcome the cold-start problem. This is only possible when there is an alternative description of the item available (for example an image or a passage of text). In Figure 2, we show how we address this problem by treating the words as indexed features, but placed within a structured feature group $\kappa_w$, the group of word features. A feature extraction function $\phi$ acts on the features in $\kappa_w$, and the other features interact with the words only via the output of $\phi$. Notice that this implies we can precompute and store the latent factors of the target task seen during training, so that predictions during inference can be sped-up. For example if we have two feature groups (e.g, a label and an item), first we compute the feature extraction

function to the unseen items and their embeddings, and then we simply apply a dot product over the stored vectors of the labels.

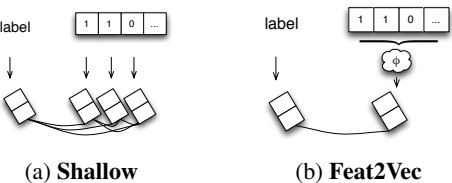

(a) **Shallow**  (b) **Feat2Vec**

Figure 2: Comparison of how factorization may use item descriptions features.

Figure 1c shows an approach of using neural networks within factorization machines that has been proposed multiple times (Dziugaite & Roy, 2015; Guo et al., 2017). It replaces the dot product of factors with a learned neural function, which has been shown to improve predictive accuracy for various tasks. In this case, fast inference for cold-start documents using pre-computed label embeddings is no longer possible. It needs to store the entire neural function that takes the embeddings as inputs. Another shortcoming of replacing the dot product with a neural function is that it would no longer be possible to interpret the embeddings as containing latent factors related to the target task; There may be highly complex mappings from the embeddings to the final output via this neural function. However, it would be straightforward to combine this approach with Feat2Vec. This is not explored in this work.

## 3.2 SUPERVISED LEARNING FROM DATA

We can learn the the parameters of a deep factorization model $\theta$ using training data by minimizing a loss function $\mathcal{L}$:

$$\arg\min_{\vec{\theta}} \sum_x \mathcal{L}\big(y(x), \hat{y}(x; \vec{\theta})\big) + \gamma ||\theta||^w \tag{5}$$

Here, $y(x)$ is the true target value for $x$ obtained from training data, and $\hat{y}(x)$ is the one estimated by the model; the hyperparameter $\gamma$ controls the amount of regularization. For the labeling and classification tasks, we optimize the binary cross-entropy for $y \in \{0, 1\}$:

$$\mathcal{L}(y, \hat{y}) = -\big(y \log(\hat{y})\big) - (1 - y) \log(1 - \hat{y})\big) \tag{6}$$

For the regression tasks where the target value is continuous, we optimize the mean squared error (MSE):

$$\mathcal{L}(y, \hat{y}) = (y - \hat{y})^2 \tag{7}$$

Neural models are typically learned using mini-batch updates, where the incremental descent is performed with respect to several instances at a time. For the implementation of this paper, we built our models using the Keras programming toolkit (Chollet et al., 2015), that is now part of Tensorflow (Abadi et al., 2015). It enables automatic differentiation, and is bundled with a general-purpose optimization algorithm called ADAM (Kingma & Ba, 2014) that needs no tuning of gradient step sizes.

It is straightforward to optimize Equation 5 directly for multiclass or binary classification. However, when the number of labels is very large, it is common practice use a binary classifier and sample the negative examples (Dyer, 2014). For the multi-label classification tasks, we use Feat2Vec with a binary output. In this case we would have at least two feature groups—one of the feature groups is the label that we want to predict, and the other group(s) is the input from which we want to make the prediction. The output indicates whether the label is associated with the input ($y = +1$), or not ($y = 0$). The datasets we use for our labeling experiments only contains positive labels, thus for each training example we sample a set of negative labels equal to the number of positive labels. It is typical to use one of the following sampling strategies according to the best validation error, in each case excluding the actual positive labels for each training example – (i) uniformly from all possible labels, or (ii) from the empirical distributions of positive labels. Other sampling strategies have been proposed (Rendle et al., 2009; Rendle & Freudenthaler, 2014).

### 3.3 Unsupervised Learning From Data

We now discuss how Feat2Vec can be used to learn embeddings in an unsupervised setting with no explicit target for prediction.

The training dataset for a Feat2Vec model consists of only the observed data. In natural language, these would be documents written by humans. Since Feat2Vec (Equation 3) requires positive and negative examples, we also need to supply unobserved data as negative examples. Consider a feature group $\vec{\kappa}_i$, that exists in very high dimensional space. For example, this could happen because we are modeling with one-hot encoding a categorical variable with large number of possible values. In such scenario, it is overwhelmingly costly to feed the model all negative labels, particularly if the model is fairly sparse.

A shortcut around this is a concept known as *implicit sampling*, where instead of using all of the possible negative labels, one simply samples a fixed number ($k$) from the set of possible negative labels for each positively labelled record. Word2Vec makes use of an algorithm called Negative Sampling, that has little theoretical guarantees (Dyer, 2014). In short, their approach samples a negative observation from a noise distribution $\mathcal{Q}_{w2v}$, that is proportional to the empirical frequency of a word in the training data.

We introduce a new implicit sampling method that enables learning unsupervised embeddings for structured feature sets. We can learn the correlation of features within a dataset by imputing negative labels, simply by generating unobserved records as our negative samples. Unlike Word2Vec, we do not constraint features types to be words. Features groups can be individual columns in a data matrix, but they need not to be. By grouping subfeatures using the parameter $\kappa$ in Equation 3, the model can reason on more abstract entities in the data. By entity, we mean a particular feature group value. For example, in our experiments on a movie dataset, we use a "genre" feature group, where we group non-mutually exclusive indicators for movie genres including comedy, action, and drama films.

We start with a dataset $S^+$ of records with $|\vec{\kappa}|$ feature groups. We then mark all observed records in the training set as positive examples. For each positive record, we generate $k$ negative labels using the following 2-step algorithm:

---

**Algorithm 1** Implicit sampling algorithm for unsupervised Feat2Vec: $\mathcal{Q}$

---

1: **function** FEAT2VEC_SAMPLE($S^+, k, \alpha_1, \alpha_2$)
2:      $S^- \leftarrow \emptyset$
3:      **for** $\vec{x}^+ \in S^+$ **do**
4:          Draw a random feature group $\kappa_i \sim \mathcal{Q}_1(\{\mathrm{params}(\phi_i)\}_{i=1}^{|\vec{\kappa}|}, \alpha_1)$
5:          **for** $j \in \{1, \dots, k\}$ **do**
6:              $\vec{x}^- \leftarrow \vec{x}^+$                                   ▷ set initially to be equal to the positive sample
7:              Draw a random feature group value $\tilde{x} \sim \mathcal{Q}_2(X_{\kappa_i}, \alpha_2)$
8:              $\vec{x}^-_{\kappa_i} \leftarrow \tilde{x}$                        ▷ substitute the $i$-th feature type with the sampled one
9:              $S^- \leftarrow S^- + \{\vec{x}^-\}$
10:         **end for**
11:     **end for**
12:     **return** $S^-$
13: **end function**

---

Explained in words, our negative sampling method for unsupervised learning iterates over all of the observations of the training dataset. For each observation $\vec{x}^+$, it randomly selects the $i$-th feature group from a noise distribution $\mathcal{Q}_1(\cdot)$. Then, it creates a negative observation that is identical to $\vec{x}^+$, except that its $i$-th feature group is replaced by a value sampled from a noise distribution $\mathcal{Q}_2(\cdot)$. In our application, we use the same class of noise distributions (flattened multinomial) for both levels of sampling, but this need not necessarily be the case.

We now describe the two noise distributions that we use. We use $P_{\mathcal{Q}}(x)$ to denote the probability of $x$ under a distribution $\mathcal{Q}$.

**Sampling Feature Groups.** The function params calculates the complexity of a feature extraction function $\phi_i$. To sample a feature group, we choose a feature group $\kappa_i$ from a multinomial distribution with probabilities proportional a feature's complexity. By complexity, we mean the number of

parameters we need to learn that are associated with a particular feature group. This choice places more weight on features that have more parameters and thus are going to require more training iterations to properly learn. The sampling probabilities of each feature group are:

$$P_{\mathcal{Q}_1}(\kappa_i|\operatorname{params}(\phi_i)\}_{i=1}^{|\vec{\kappa}|}, \alpha_1) \; = \frac{\operatorname{params}(\phi_i)^{\alpha_1}}{\sum_{j=1}^{|\vec{\kappa}|}\operatorname{params}(\phi_j)^{\alpha_1}}, \quad \alpha_1 \in [0,1] \tag{8}$$

For categorical variables using a linear fully-connected layer, the complexity is simply proportional to the number of categories in the feature group. However, if we have multiple intermediate layers for some feature extraction functions (e.g., convolutional layers), these parameters should also be counted towards a feature group's complexity. The hyper-parameter $\alpha_1$ helps flatten the distribution. When $\alpha_1 = 0$, the feature groups are sampled uniformly, and when $\alpha_1 = 1$, they are sampled proportional to their complexity. Figure A.1 in the Appendix provides a visualization of how the feature sampling rate varies with the hyperparameter for features with differing levels of complexity.

**Sampling Feature Group Values.** To sample a value from within a feature groups $\kappa_i$, we use a similar strategy to Word2Vec and use the empirical distribution of values:

$$P_{\mathcal{Q}_2}(x|X_{\kappa_i}, \alpha_2) \; = \frac{\operatorname{count}(x)^{\alpha_2}}{\sum_{x'_{\kappa_i} \in S^+} \operatorname{count}(x'_{\kappa_i})^{\alpha_2}}, \quad \alpha_2 \in [0,1] \tag{9}$$

Here, $\operatorname{count}(x)$ is the number of times a feature group value $x$ appeared in the training dataset $S^+$, and $\alpha_2$ is again a flattening hyperparameter.

This method will sometimes by chance generate negatively labeled samples that *do* exist in our sample of observed records. The literature offers two possibilities: in the Negative Sampling that Word2Vec follows, the duplicate negative samples are simply ignored (Dyer, 2014). Alternatively, it is possible to account for the probability of random negative labels that are identical to positively labeled data using Noise Contrastive Estimation (NCE) (Gutmann & Hyvärinen, 2010).

### 3.3.1 THE LOSS FUNCTION FOR UNSUPERVISED LEARNING

For our unsupervised learning of embeddings, we optimize a NCE loss function, to adjust the structural statistical model $\hat{y} = p(y = 1|\vec{x}, \vec{\phi}, \theta)$, expressed in Equation 3 to account for the possibility of random negative labels that appear identical to positively labeled data. $\theta$ here represents the parameters learned in during training (i.e. the $b_i$ terms and parameters associated with the extraction functions $\phi_i$ in Equation 3). Since we only deal with a dichotomous label, indicating a positive or negative sample, for unsupervised learning, we restrict our attention to usage of Equation 3 with $\omega$ as a logistic link function.

An additional burden of NCE is that we need to calculate a partition function $Z_{\vec{x}}$ for each unique record type $\vec{x}$ in the data that transforms the probability $\hat{y}$ of a positive or negative label into a well-behaved distribution that integrates to 1. Normally, this would introduce an astronomical amount of computation and greatly increase the complexity of the model. As a work-around, we appeal to the work of Mnih & Teh (2012), who showed that in the context of language models that setting the $Z_{\vec{x}} = 1$ in advance effectively does not change the performance of the model. The intuition is that if the underlying model has enough free parameters that it will effectively learn the probabilities itself. Thus, it does not over/under predict the probabilities on average (since that will result in penalties on the loss function).

Written explicitly, the new structural probability model is:

$$\tilde{p}(Y = 1|\vec{x}, \vec{\phi}, \theta) = \frac{\exp\big(s(\vec{x}, \vec{\phi}, \theta)\big)}{\exp\big(s(\vec{x}, \vec{\phi}, \theta)\big) + P_{\mathcal{Q}}(\vec{x}|\alpha_1, \alpha_2)} \tag{10}$$

where $s(.)$ denotes the score of a record $\vec{x}$ given parameter values/extraction functions:

$$s(\vec{x}, \vec{\phi}, \theta) = b_0 + \sum_{i=1}^{n} b_i x_i \; + \sum_{i=1}^{|\vec{\kappa}|}\sum_{j=i}^{|\vec{\kappa}|} \phi_i(\vec{x}_{\vec{\kappa}_i}) \cdot \phi_j(\vec{x}_{\vec{\kappa}_j}) \tag{11}$$

and $P_\mathcal{Q}(.)$ denotes the total probability of a record $\vec{x_i}$ being drawn from our negative sampling algorithm, conditional on the positively labeled record $\vec{x}^+$ the negative sample is drawn for:

$$P_\mathcal{Q}(\vec{x}|\alpha_1, \alpha_2, X, \vec{x}^+) = P_{\mathcal{Q}_2}(\vec{x}_{\kappa_i}|X_{\kappa_i}, \alpha_2)P_{\mathcal{Q}_1}(\kappa_i| \text{params}(\phi_i))\}_{i=1}^n, \alpha_1) \quad (12)$$

Our loss function $L$ optimizes $\theta$, the parameters of the feature extraction functions $\vec{\phi}$, while accounting for the probability of negative samples.

$$L(S) = \arg\min_\theta \frac{1}{|S^+|} \sum_{\vec{x}^+ \in S^+} \left( \log(\tilde{p}(y=1|\vec{x}^+, \vec{\phi}, \theta)) + \sum_{\vec{x}^- \sim \mathcal{Q}(\cdot|\vec{x}^+)}^k \log(\tilde{p}(y=0|\vec{x}^-, \vec{\phi}, \theta)) \right)$$
$$(13)$$

Feat2Vec has interesting theoretical properties. For example, it is well known that Factorization Machines can be used as a multi-label classifier: with at least two features, one can use one of the feature as the target label, and the other as the input feature to make a prediction. In such setting, the output indicates whether the label is associated with the input ($y = +1$), or not ($y = 0$), and therefore the input can be associated with more than one label. With $n$ feature types, Feat2Vec is equivalent to optimizing a convex combination of the loss functions from $n$ individual Factorization Machines. In other words, it optimizes $n$ multi-label classifiers, where each classifier is optimized for a different target (i.e.,a specific feature group). We show the proof of this in the Appendix 1.

## 4 EMPIRICAL RESULTS

### 4.1 SUPERVISED EMBEDDINGS

We now address our working hypotheses for evaluating supervised embeddings. For all our experiments we define a development set and a single test set which is 10% of the dataset, and a part of the development set is used for early stopping or validating hyper-parameters. Since these datasets are large and require significant time to train on an Nvidia K80 GPU cluster, we report results on only a single training-test split. For the multi-label classification task in 4.1.1 we predict a probability for each document-label pair and use an evaluation metric called Area Under the Curve (AUC) of the Receiver Operating Characteristic (ROC). Since we only observe positive labels, for each positive label in the test set we sample negative labels according to the label frequency. This ensures that if a model merely predicts the labels according to their popularity, it would have an AUC of 0.5. A caveat of our evaluation strategy is that we could be underestimating the performance of our models—there is a small probability that the sampled negatives labels are false negatives. However, since we apply the same evaluation strategy consistently across our methods and baselines, the relative difference of the AUC is meaningful. We choose the AUC as a metric because it is popular for both classification and ranking problems. For the regression task in 4.1.2, we use mean squared error (MSE) as the evaluation metric. In preliminary experiments we noticed that regularization slows down convergence with no gains in prediction accuracy, so we avoid overfitting only by using early stopping. We share most of the code for the experiments online[1] for reproducibility.

For our feature extraction function $\phi$ for text, we use a Convolutional Neural Network (CNN) that has been shown to be effective for natural language tasks (Kalchbrenner et al., 2014; Weston et al., 2014). In Appendix B we describe this network and its hyper-parameters. Instead of tuning the hyper-parameters, we follow previously published guidelines (Zhang & Wallace, 2015).

### 4.1.1 IS Feat2Vec EFFECTIVE FOR COLD-START PREDICTIONS?

We compare Feat2Vec with an extension of matrix factorization that can generalize to unseen items for text documents, Collaborative Topic Regression (CTR– Wang & Blei (2011)), a method with an open-source Python implementation[2]. We evaluate them on the **CiteULike** dataset which consists of pairs of scientific articles and the users who have added them to their personal libraries, and it contains 16,980 unique articles and 5,551 unique users. We use the models to predict users who

---

[1] https://goo.gl/zEQBiA
[2] https://github.com/arongdari/python-topic-model

Table 1: Yelp rating prediction

|  | MSE | Improvement over Matrix Factorization |
| --- | --- | --- |
| Matrix Factorization | 1.561 | - |
| Feat2Vec | **0.480** | **69.2** % |
| DeepCoNN | 1.441 | 19.6 % |

may have added a given article to their library. We compare the performance of Feat2Vec with CTR using pre-defined cross-validation splits[3]. We use 1% of the training set for early stopping.

For CTR we use the hyper-parameters reported by the authors as best, except for $r$ which we found had a significant impact on training time . We only consider $r \in \{5, 10, 15\}$ and choose the value which gives the best performance for CTR (details in Appendix A.2). On the warm-start condition, CTR has an AUC of 0.9356; however, it shows significant degradation in performance for unseen documents and it only performs slightly better than random chance with an AUC of 0.5047. On the other hand, Feat2Vec achieves AUC of 0.9401 on the warm-start condition, and it only degrades to 0.9124 on unseen documents. Feat2Vec can also be trained over ten times faster, since it can leverage GPUs.[4] We also note that we have not tuned the architecture or hyper-parameters of the feature extraction function $\phi$ and greater improvements are possible by optimizing them.

#### 4.1.2 COMPARISON WITH ALTERNATIVE CNN-BASED TEXT FACTORIZATION

We now compare with a method called DeepCoNN, a deep network specifically designed for incorporating text into matrix factorization (Zheng et al., 2017)—which reportedly, is the state of the art for predicting customer ratings when textual reviews are available. For Feat2Vec we use the same feature extraction function (see Appendix B.1 for details) used by DeepCoNN. We evaluate on the Yelp dataset[5], which consists of 4.7 million reviews of restaurants. For each user-item pair, DeepCoNN concatenates the text from all reviews for that item and all reviews by that user. The concatenated text is fed into a feature extraction function followed by a factorization machine. In contrast, for Feat2Vec, we build 3 feature groups: item identifiers (in this case, restaurants), users and review text.

Table 1 compares our methods to DeepCoNN's published results because a public implementation is not available. We see that Feat2Vec provides a large performance increase when comparing the reported improvement, over Matrix Factorization, of the mean squared error. Our approach is more general, and we claim that it is also more efficient. Since DeepCoNN concatenates text, when the average reviews per user is $\bar{n}_u$ and reviews per item is $\bar{n}_i$, each text is duplicated on average $\bar{n}_i \times \bar{n}_u$ times per training epoch. In contrast, for Feat2Vec each review is seen only once per epoch. Thus it can be 1-2 orders of magnitude more efficient for datasets where $\bar{n}_i \times \bar{n}_u$ is large.

### 4.2 GENERAL-PURPOSE EMBEDDINGS

#### 4.2.1 DOES Feat2Vec ENABLE BETTER EMBEDDINGS?

Ex ante, it is unclear to us how to evaluate the performance of an unsupervised embedding algorithm, but we felt that a reasonable task would be a ranking task one might practically attempt using our datasets. This task will assess the similarity of trained embeddings using unseen records in a left-out dataset. In order to test the relative performance of our learned embeddings, we train our unsupervised Feat2Vec algorithm and compare its performance in a targeted ranking task to Word2Vec's CBOW algorithm for learning embeddings. In our evaluation approach, we compare the cosine similarity of the embeddings of two entities where these entities are known to be associated with each other since they appear in the same observation in a test dataset. In particular, in the movie dataset

---

[3]For warm-start we use `https://www.cs.cmu.edu/~chongw/data/citeulike/folds/cf-train-1-items.dat` and for cold-start predictions, we use the file `ofm-train-1-items.dat` and the corresponding test sets for each

[4]Feat2Vec and MF were trained on an Nvidia K80 GPU, while CTR was trained on a Xeon E5-2666 v3 CPU.

[5]https://www.yelp.com/dataset/challenge

we compare the similarity of movie directors to those of actors who were cast in the same film for a left-out set of films. For our educational dataset, we compare rankings of textbooks by evaluating the similarity of textbook and user embeddings. We evaluate the rankings according to their mean percentile rank (MPR):

$$MPR = \frac{1}{N} \sum_{i=1}^{N} \frac{R_i}{\max R}$$

where $R_i$ is the rank of the entity under our evaluation procedure for observation $i$. This measures on average how well we rank actual entities. A score of 0 would indicate perfect performance (i.e. top rank every test sample given), so a lower value is better under this metric. See the appendix §A.1 for further details on the experimental setup.

### 4.2.2 DATASETS

**Movies** The Internet Movie Database (IMDB) is a publicly available dataset[6] of information related to films, television programs and video games. Though in this paper, we focus only on data on its 465,136 movies. Table A.1 in the appendix (§A.1) summarizes the feature types we use. It contains information on writers, directors, and principal cast members attached to each film, along with metadata.

**Education** We use a dataset from an anonymized leading technology company that provides educational services. In this proprietary dataset, we have 57 million observations and 9 categorical feature types which include textbook identifier, user identifier, school identifier, and course the book is typically used with, along with other proprietary features. Here, each observation is an "interaction" a user had with a textbook.

### 4.2.3 RESULTS

After training, we use the cast members associated with the movies of the test set and attempt to predict the actual director the film was directed. We take the sum of the cast member embeddings, and rank the directors by cosine similarity of their embeddings to the summed cast member vector. If there is a cast member in the test dataset who did not appear in the training data, we exclude them from the summation. For the educational dataset, we simply use the user embedding directly to get the most similar textbooks.

Table 2 presents the results from our evaluation. Feat2Vec sizably outperforms CBOW in the MPR metric. In fact, Feat2Vec predicts the actual director 2.43% of the times, while CBOW only does so 1.26% of the time, making our approach almost 2 times better in terms of Top-1 Precision metric. We explore in greater detail the distribution of the rankings in the appendix in §A.2.

Table 2: Mean percentile rank

| Dataset | Feat2Vec | CBOW |
|---|---|---|
| IMDB | 19.36% | 24.15% |
| Educational | 25.2% | 29.2% |

### 4.3 UNSUPERVISED Feat2Vec PERFORMANCE WITH CONTINUOUS INPUTS

We now focus on how well Feat2Vec performs on a real-valued feature with a complex feature extraction function. We expect this task to highlight Feat2Vec's advantage over token-based embedding learning algorithms, such as Word2Vec, since our rating embedding extraction function will require embeddings of numerically similar ratings to be close , while Word2Vec will treat two differing ratings tokens as completely different entities. We evaluate the prediction of the real-valued rating of movies in the test dataset by choosing the IMDB rating embedding most similar[7] to the embedding of the movie's director, and compute the Root Mean Squared Error (RMSE) of the predicted rating in the test dataset. We also vary $\alpha_1$, the flattening hyperparameter for feature group

---

[6]http://www.imdb.com/interfaces/
[7]As before, the metric is cosine similarity.

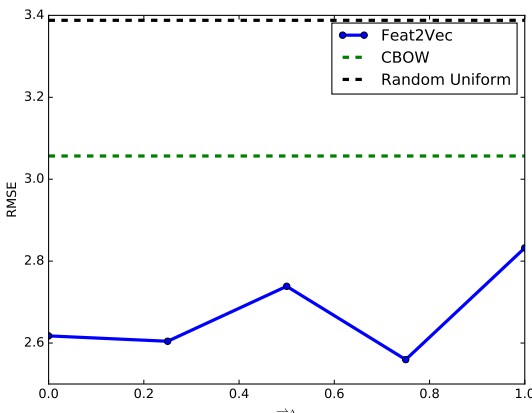

Figure 3: RMSE in Ratings Task as a Function of $\alpha_1$

sampling, to see what effect this hyperparameter has on our performance. Intuitively, a low $\alpha_1$ will greatly improve the quality of the ratings embeddings learned, since it has relatively few parameters and is otherwise sampled infrequently. At the same time, with low $\alpha_1$ the director feature will be sampled less since it is one of the most complex features to learn, so the learned director embeddings may be of poorer quality. Figure 3 displays the results of our experiment, benchmarked against the performance of Word2Vec's CBOW algorithm in the prediction task. We also show as a baseline the RMSE of a random uniform variable over the range of possible ratings (0 to 10). As is evident from the plot, CBOW performs a bit better than a random prediction, but is also handily outperformed by Feat2Vec across all hyper-parameter settings. The algorithm's performance does not seem very sensitive to the hyperparameter choice.

## 5   RELATION TO PRIOR WORK

The original Factorization Machine formulation has been extended for multiple contexts. For example, Field-Aware Factrorization Machine (Juan et al., 2016) allows different weights for some feature interactions, but does not allow feature groups or feature extraction functions like Feat2Vec does.

Algorithms that calculate continuous representations of entities other than words have been proposed for biological sequences (Abrahamsson & Plotkin, 2009), of vertices in network graphs (Perozzi et al., 2014) or in machine translation for embeddings of complete sentences (Kiros et al., 2015). Generative Adversarial Networks (Goodfellow et al., 2014)(GANs) have been used to produce unsupervised embeddings of images effective for classification (Radford et al., 2015) and for generating natural language (Press et al., 2017). To our knowledge, GANs have not been used for jointly embedding multiple feature types. Adversarial training could be an alternative to NCE for unsupervised learning, but we leave this for future study.

We recently discovered a promising direction for an algorithm still in development called StarSpace (Wu et al., 2017) with similar goals from ours. Even though they intend to be able to embed all types of features, at the time of the writing of this paper, their pre-print method was limited to only work for bag of words. While Feat2Vec can jointly learn embeddings for all feature values in a dataset, StarSpace samples a single arbitrary feature. Our preliminary experiments suggest that sampling a single feature does not produce embeddings that generalize well. Nonetheless, a limitation of our work is that we do not compare with StarSpace, which future work may decide to do.

## 6 CONCLUSION

Embeddings have proven useful in a wide variety of contexts, but they are typically built from datasets with a single feature type as in the case of Word2Vec, or tuned for a single prediction task as in the case of Factorization Machine. We believe Feat2Vec is an important step towards general-purpose methods, because it decouples feature extraction from prediction for datasets with multiple feature types, it is general-purpose, and its embeddings are easily interpretable.

In the supervised setting, Feat2Vec is able to calculate embeddings for whole passages of texts, and we show experimental results outperforming an algorithm specifically designed for text—even when using the same feature extraction CNN. This suggests that the need for ad-hoc networks should be situated in relationship to the improvements over a general-purpose method.

In the unsupervised setting, Feat2Vec's embeddings are able to capture relationships across features that can be twice as better as Word2Vec's CBOW algorithm on some evaluation metrics. Feat2Vec exploits the structure of a datasets to learn embeddings in a way that is structurally more sensible than existing methods. The sampling method, and loss function that we use have interesting theoretical properties. To the extent of our knowledge, Unsupervised Feat2Vec is the first method able to calculate continuous representations of data with arbitrary feature types.

Future work could study how to reduce the amount of human knowledge our approach requires; for example by automatically grouping features into entities, or by automatically choosing a feature extraction function. These ideas can extend to our codebase that we make available [8]. Overall, we evaluate supervised and unsupervised Feat2Vec on 2 datasets each. Though further experimentation is necessary, we believe that our results are an encouraging step towards general-purpose embedding models.

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

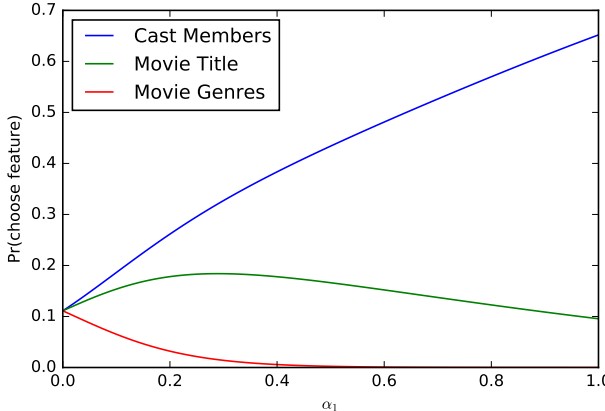

Figure A.1: Feature Sampling Probabilities as a Function of $\alpha_1$

Table A.1: IMDB dataset features

| Feature Type Name | Type | # of feats. | Example for an instance |
|---|---|---|---|
| Runtime (minutes) | Real-valued | 1 | 116 |
| IMDB rating (0-10) | Real-valued | 1 | 7.8 |
| # of IMDB rating votes | Real-valued | 1 | 435,682 |
| Is adult film? | Boolean | 2 | False |
| Movie releaes year | Categorical | 271 | 2001 |
| Movie title | Text | 165,471 | "Ocean's", "Eleven" |
| Directors | Bag of categories | 174,382 | 'Steven Soderbergh" |
| Genres | Bag of categories | 28 | "Crime", "Thriller" |
| Writers | Bag of categories | 244,241 | "George Johnson", "Jack Russell" |
| Principal cast members (actors) | Bag of categories | 1,104,280 | "George Clooney", "Brad Pitt", "Julia Roberts" |

# A  APPENDIXES

## A.1  UNSUPERVISED RANKING EXPERIMENT DETAILS

For our evaluation, we define a testing set that was not used to tune the parameters of the model. For the IMDB dataset, we randomly select a 10% sample of the observations that contain a director that appears at least twice in the database [9]. We do this to guarantee that the set of directors in the left-out dataset appear during training at least once, so that each respective algorithm can learn something about the characteristics of these directors. For the educational dataset, our testing set only has observations of textbooks and users that appear at least 10 times in training.

For both Feat2Vec and CBOW, we perform cross-validation on the loss function, by splitting the 10% of the training data randomly into a validation set, to determine the number of epochs to train, and then train the full training dataset with this number of epochs. [10] While regularization of the embeddings during training is possible, this did not dramatically change results, so we ignore this dimension of hyperparameters.

We rank left-out entity pairs in the test dataset using the ordinal ranking of the cosine similarity of target and input embeddings. For the IMDB dataset, the target is the director embedding, and the

---

[9]Over 90% of the movies in the database have exactly one director, but in cases where there are multiple directors to a film, we use the first director listed in the IMDB dataset.

[10]Because we train the Word2Vec CBOW algorithm with the `gensim` python library, it is impossible to recover the output weight matrix, and so the loss function is inaccessible for an outside document set. So, we created our own loss function that measures average within-document cosine similarity of all possible token pairs. The result was that both algorithms are trained for a similar number of epochs.

input embedding is the sum of the cast member embeddings. For the educational dataset, the target is the textbook embedding, and the input embedding is the user embedding.

For training Feat2Vec we set $\alpha_1 = \alpha_2 = 3/4$ in the IMDB dataset; and $\alpha_1 = 0$ and $\alpha_2 = 0.5$ for the educational. In each setting, $\alpha_2$ is set to the same flattening hyperparameter we use for CBOW to negatively sample words in a document. We learn $r = 50$ dimensional embeddings under both algorithms.

Below we describe how CBOW is implemented on our datasets for unsupervised experiments and what extraction functions are used to represent features in the IMDB dataset.

**Word2Vec** For every observation in each of the datasets, we create a document that tokenizes the same information that we feed into Feat2Vec. We prepend each feature value by its feature name, and we remove spaces from within features. In Figure A.2 we show an example document. Some features may allow multiple values (e.g., multiple writers, directors). To feed these features into the models, for convenience, we constraint the number of values, by truncating each feature to no more than 10 levels (and sometimes less if reasonable). This results in retaining the full set of information for well over 95% of the values. We pad the sequences with a "null" category whenever necessary to maintain a fixed length. We do this consistently for both Word2Vec and Feat2Vec. We use the CBOW Word2Vec algorithm and set the context window to encompass all other tokens in a document during training, since the text in this application is unordered.

Runtime_116, IMDB_rating_7.8, ..., Writers_George_Johnson, Writers_Jack_Russell

Figure A.2: Sample document for Word2Vec for the Ocean's Eleven movie

**Feat2Vec** Feature representation in Feat2Vec requires a feature extraction function for each feature type. Here, we explain how we build these functions:

- **Bag of categories, categorical, and boolean:** For all of the categorical variables, we learn a unique $r$-dimensional embedding for each entity using a linear fully-connected layer (Equation 4). We do not require one-hot encodings, and thus we allow multiple categories to be active; resulting in a single embedding for the group that is the sum of the embeddings of the subfeatures. This is ordering-invariant: the embedding of "Brad Pitt" would be the same when he appears in a movie as a principal cast member, regardless whether he was 1st or 2nd star. Though, if he were listed as a director it may result in a different embedding.

- **Text:** We preprocess the text by removing non alpha-numeric characters, stopwords, and stemming the remaining words. We then follow the same approach that we did for categorical variables, summing learned word embeddings to a "title embedding" before interacting. It would be easy to use more sophisticated methods (e.g, convolutions), but we felt this would not extract further information.

- **Real-valued:** For all real-valued features, we pass these features through a 3-layer feedforward fully connected neural network that outputs a vector of dimension $r$, which we treat as the feature's embedding. Each intermediate layer has $r$ units with `relu` activation functions. These real-valued features highlight one of the advantages of the Feat2Vec algorithm: using a numeric value as an input, Feat2Vec can learn a highly nonlinear relation mapping a real number to our high-dimensional embedding space. In contrast, Word2Vec would be unable to know ex ante that an IMDB rating of 5.5 is similar to 5.6.

## A.2 DISTRIBUTION OF IMDB DIRECTOR RANKINGS

Figure A.3 shows the full distribution of rankings of the IMDB dataset, rather than summary statistics, in the form of a Cumulative Distribution Function (CDF) of all rankings calculated in the test dataset. The graphic makes it apparent for the vast majority of the ranking space, the rank CDF of Feat2Vec is to the left of CBOW, indicating a greater probability of a lower ranking under Feat2Vec. This is not, however, the case at the upper tail of ranking space, where it appears CBOW is superior.

However, when we zoom-in on the absolute upper region of rankings (1 to 25), which might be a sensible length of ranks one might give as actual recommendatiosn, it is the case that up until rank 8 or so, Feat2Vec outperforms CBOW still. Intermediate rankings are still strong signals that our Feat2Vec algorithm is doing a better job of extracting information into embeddings, particularly those entities that appear sparsely in the training data and so are especially difficult to learn.

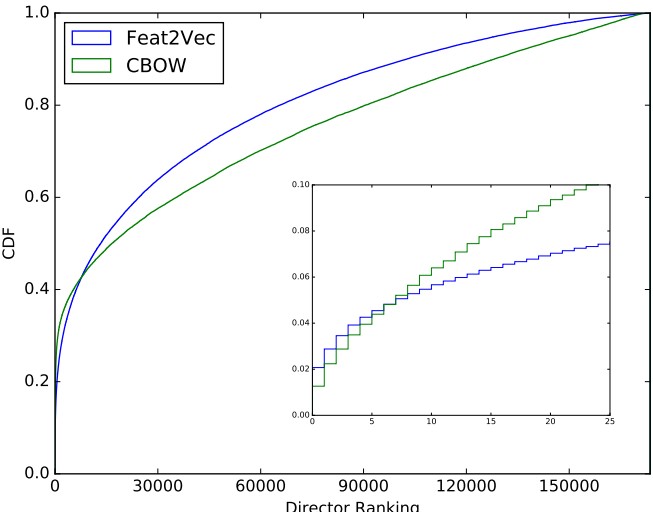

Figure A.3: Cumulative Distribution Function of Director Rankings
(With Zoom-in to Top 25 Ranks)

## A.3 Proof to Theorem 1

**Theorem 1.** *The gradient for learning embeddings with* Feat2Vec *is a convex combination of the gradient from* $n$ *targeted Factorization Machines for each feature in the data when each feature group is a singleton, where* $n$ *is the total number of features in the dataset.*

*Proof.* Let $S^+_{\kappa_i}$ denote the positively labeled records whose corresponding negative samples resample feature $\kappa_i$. For convenience, suppress the inclusion of learned parameters $\theta$ in the notation in this section while understanding the feature extraction functions $\vec{\phi}$ implicitly include these parameters. We can express the loss function $L(.)$, the binary cross-entropy of the data given the Feat2Vec model, as follows:

$$L(S^+|\vec{\phi}) = \frac{1}{|S^+|} \sum_{\vec{x}^+ \in S^+} \Big( \log(\tilde{p}(y=1|\vec{\phi}, \vec{x}^+)) + \sum_{\vec{x}^- \sim \mathcal{Q}(.|\vec{x}^+)}^{k} \log(\tilde{p}(y=0|\vec{\phi}, \vec{x}^-)) \Big)$$

$$= \frac{1}{|S^+|} \sum_{\vec{x}^+ \in S^+} \Big( \log(\tilde{p}(y=1|\vec{\phi}, \vec{x}^+, \vec{x}^+ \in S_{\kappa_i}^+)p(\vec{x}^+ \in S_{\kappa_i}^+))$$

$$+ \sum_{\vec{x}^- \sim \mathcal{Q}(.|\vec{x}^+)}^{k} \log(\tilde{p}(y=0|\vec{\phi}, \vec{x}^-, \vec{x}^+ \in S_{\kappa_i}^+)p(\vec{x}^+ \in S_{\kappa_i}^+)) \Big)$$

$$= \frac{1}{|S^+|} \sum_{i=1}^{n} \sum_{\vec{x}^+ \in S_{\kappa_i}^+} \Big( \log(\frac{e^{s(\vec{x}^+, \vec{\phi})}p(\vec{x}^+ \in S_{\kappa_i}^+)}{e^{s(\vec{x}^+, \vec{\phi})} + P_\mathcal{Q}(\vec{x}^+|\vec{x}^+, \vec{x}^+ \in S_{\kappa_i}^+)})$$

$$+ \sum_{\vec{x}^- \sim \mathcal{Q}(.|\vec{x}^+, \vec{x}^+ \in S_{\kappa_i}^+)}^{k} \log(\frac{P_\mathcal{Q}(\vec{x}^-|\vec{x}^+, \vec{x}^+ \in S_{\kappa_i}^+)p(\vec{x}^+ \in S_{\kappa_i}^+)}{e^{s(\vec{x}^-, \vec{\phi})} + P_\mathcal{Q}(\vec{x}^-|\vec{x}^+, \vec{x}^+ \in S_{\kappa_i}^+)}) \Big)$$

Note now that $P_\mathcal{Q}(\vec{x}|\vec{x}^+, \vec{x}^+ \in S_{\kappa_i}^+)$ is simply the probability of the record's feature value $\vec{x}_f$ under the second step noise distribution $\mathcal{Q}_2(X_f, \alpha_2)$: $P_\mathcal{Q}(\vec{x}|\vec{x}^+, \vec{x}^+ \in S_{\kappa_i}^+) = P_{\mathcal{Q}_2}(\vec{x}_f)$

$$= \frac{1}{|S^+|} \sum_{i=1}^{n} \sum_{\vec{x}^+ \in S_{\kappa_i}^+} \Big( \log(\frac{e^{s(\vec{x}^+, \vec{\phi})}p(\vec{x}^+ \in S_{\kappa_i}^+)}{e^{s(\vec{x}^+, \vec{\phi})} + P_{\mathcal{Q}_2}(\vec{x}_{\kappa_i}^+)}) + \sum_{\vec{x}^- \sim \mathcal{Q}(.|\vec{x}^+, i \in S_{\kappa_i}^+)}^{k} \log(\frac{P_{\mathcal{Q}_2}(\vec{x}_f^-)p(\vec{x}^+ \in S_{\kappa_i}^+)}{e^{s(\vec{x}^-, \vec{\phi})} + P_{\mathcal{Q}_2}(\vec{x}_f^-)}) \Big)$$

$$= \frac{1}{|S^+|} \sum_{i=1}^{n} \sum_{\vec{x}^+ \in S_{\kappa_i}^+} \Big( \log(\frac{e^{s(\vec{x}^+, \vec{\phi})}}{e^{s(\vec{x}^+, \vec{\phi})} + P_{\mathcal{Q}_2}(\vec{x}_{\kappa_i}^+)}) + \log(p(\vec{x}^+ \in S_{\kappa_i}^+)^{k+1})$$

$$+ \sum_{\vec{x}^- \sim \mathcal{Q}(.|\vec{x}^+, \vec{x}^+ \in S_{\kappa_i}^+)}^{k} \log(\frac{P_{\mathcal{Q}_2}(\vec{x}_f^-)}{e^{s(\vec{x}^-, \vec{\phi})} + P_{\mathcal{Q}_2}(\vec{x}_f^-)}) \Big)$$

We now drop the term containing the probability of assignment to a feature group $p(\vec{x}^+ \in S_{\kappa_i}^+)$ since it is outside of the learned model parameters $\vec{\phi}$ and fixed in advance:

$$\propto \frac{1}{|S^+|} \sum_{i=1}^{n} \sum_{\vec{x}^+ \in S_{\kappa_i}^+} \Big( \log(\frac{e^{s(\vec{x}^+, \vec{\phi})}}{e^{s(\vec{x}^+, \vec{\phi})} + P_{\mathcal{Q}_2}(\vec{x}_{\kappa_i}^+)}) + \sum_{\vec{x}^- \sim \mathcal{Q}(.|\vec{x}^+, \vec{x}^+ \in S_{\kappa_i}^+)}^{k} \log(\frac{P_{\mathcal{Q}_2}(\vec{x}_f^-)}{e^{s(\vec{x}^-, \vec{\phi})} + P_{\mathcal{Q}_2}(\vec{x}_f^-)}) \Big)$$

$$\xrightarrow[|S^+| \to \infty]{} \sum_{i=1}^{n} p(\vec{x}^+ \in S_{\kappa_i}^+) E\Big[ \log(\frac{e^{s(\vec{x}^+, \vec{\phi})}}{e^{s(\vec{x}^+, \vec{\phi})} + P_{\mathcal{Q}_2}(\vec{x}_{\kappa_i}^+)}) + \sum_{\vec{x}^- \sim \mathcal{Q}(.|\vec{x}^+, \vec{x}^+ \in S_{\kappa_i}^+)}^{k} \log(\frac{P_{\mathcal{Q}_2}(\vec{x}_f^-)}{e^{s(\vec{x}^-, \vec{\phi})} + P_{\mathcal{Q}_2}(\vec{x}_f^-)}) \Big]$$

$$= \sum_{i=1}^{n} p(\vec{x}^+ \in S_{\kappa_i}^+) E\Big[ L(\vec{x}|\vec{\phi}, \text{target} = f) \Big]$$

Thus, the loss function is just a convex combination of the loss functions of the targeted classifiers for each of the $p$ features, and by extension so is the gradient since:

$$\frac{\partial}{\partial \phi} \sum_{i=1}^{n} p(\vec{x}^+ \in S_{\kappa_i}^+) E\Big[ L(\vec{x}|\vec{\phi}, \text{target} = f) \Big] = \sum_{i=1}^{n} p(\vec{x}^+ \in S_{\kappa_i}^+) \frac{\partial}{\partial \phi} E\Big[ L(\vec{x}|\vec{\phi}, \text{target} = f) \Big]$$

Thus the algorithm will, at each step, learn a convex combination of the gradient for a targeted classifier on feature $f$, with weights proportional to the feature group sampling probabilities in step 1 of

the sampling algorithm. Note that if feature groups are not singletons, the gradient from unsupervised Feat2Vec will analogously be a convex combination of $n$ gradients learned from supervised learning tasks on each of the $n$ feature groups. □

## B   FEATURE EXTRACTION NETWORK FOR NATURAL LANGUAGE

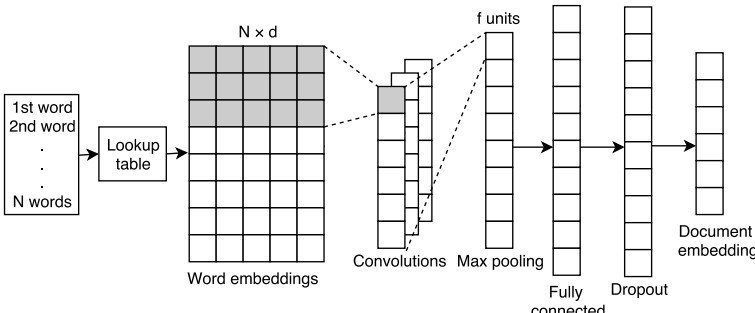

Figure A.4: Feature extraction network used for labelling tasks. We use f=1000 convolutional filters each of width 3 (words)

Here we describe the details of the feature extraction function $\phi$ used in our experiments for supervised tasks in §4.1. An overview of the network is given in Fig. A.4. We choose the most common words of each dataset to build a vocabulary of size $n$, and convert the words of each document to a sequence of length $t$ of one-hot encodings of the input words. If the input text is shorter than $t$, then we pad it with zeros; if the text is longer, we truncate it by discarding the trailing words. Therefore, for a vocabulary size $n$, the input has dimensions $t \times n$. These $t\times$ dimensional matrix is then passed through the following layers:

1. We use an embedding layer to assign a $d$-dimensional vector to each word in the input passage of text. This is done through a $d \times n$-dimensional lookup table, which results in an $t \times d$ matrix.

2. We extract features from the embeddings with functions called *convolutional filters* (LeCun et al., 1998) (also called feature maps). A convolutional filter is simply a matrix learned from an input. We learn $f$ filters that are applied on groups of $m$ adjacent word embeddings, thus each of our filters is a $d \times m$ matrix of learned parameters. Filters are applied by computing the element-wise dot product of the filter along a sliding window of the entire input. The resulting output for each filter is a vector of length $t - m + 1$. We also apply a ReLU activation to the output of each filter.

3. Consider the case of inputs of different lengths. For very short texts, the output of the filters will be mostly zero since the input is zero-padded. To enforce learning from the features of the text, and not just its length we apply a function called 1-max pooling to the output of the filters: from the $t - m + 1$ output vector of each filter, we select the maximum value. This yields a vector of length $F$, a representation of the passage which is independent of its length.

4. We learn higher-level features from the convolutional filters. For this, we use a fully connected layer with $p$ units and a ReLU activation,

5. During training (not in inference), we prevent the units from co-adapting too much with a dropout layer (Srivastava et al., 2014). Dropout is a form of regularization that for each mini-batch randomly drops a specified percentage of units.

6. the final embedding for $x_j$ (that is used in the factorization) is computed by a dense layer with $r$ output units and an activation function, where $r$ is the embedding size of our indexable items.

We set the maximum vocabulary size $n$ to 100,000 words, and input embedding size $d$ to 50 for all experiments. We initialize the input word embeddings and the label embeddings using

Word2Vec(Mikolov et al., 2013) We have have not evaluated multiple architectures or hyper-parameter settings and obtain good results on diverse datasets with the same architecture, which was designed followed recommendations from a large scale evaluation of CNN hyper parameters(Zhang & Wallace, 2015). We set the number of convolutional filters $f$ to 1,000, and the dropout rate to 0.1. The maximum sequence length $t$ was chosen according to the typical document length (350 words for CiteULike and 250 for Yelp). For the CTR dataset, because we use very small values of $r$, due to the tendency of the ReLU units to 'die' during training (output zero for all examples), which can have a significant impact, we used instead PReLU activations (He et al., 2015) for the final layer, since they do not suffer from this issue.

### B.1 Feature Extraction for DeepCoNN comparison

The CNN architecture used for DeepCoNN (Zheng et al., 2017) is similar to the previous section. It consists of a word embedding lookup table, convolutional layer, 1-max pooling and a fully connected layer. We use the hyper-parameters that the authors report as best - 100 convolution filters and 50 units for the fully connected layer. We set the word embedding size to 100, the vocabulary size to 100,000 and the maximum document length to 250.

## C Hyper-parameters for CTR

To compare Feat2Vec with Collaborative Topic Regression, we choose the embedding size $r \in \{5, 10, 15\}$ for which CTR performs best. The results are show in Table A.2.

Table A.2: Tuning embedding size for CTR

|              | r=5    | r=10   | r=15   | Time (mins.) |
|--------------|--------|--------|--------|--------------|
| Matrix Fact. | 0.8723 | 0.8911 | 0.9046 | **1**        |
| Feat2Vec     | **0.9081** | **0.9303** | **0.9401** | 133      |
| C.T.R        | 0.8763 | 0.9234 | 0.9356 | 1425         |

