# OpenReview forum: "Feat2Vec:  Dense Vector Representation for Data with Arbitrary Features"
_ICLR.cc/2018/Conference — Reject_

### Official Review · AnonReviewer2 · 2017-11-24
**Interesting paper with convincing results, but the approach has limited novelty. Proposed method is based on an approach that is concurrently under review at ICLR18**

**Rating:** 7
**Confidence:** 3

**Review:**

Summary:
This paper proposes an approach to learn embeddings for structured datasets i.e. datasets which have heterogeneous set of features, as opposed to just words or just pixels. The paper proposes an approach called Feat2vec that relies on Structured Deep-In Factorization machines-- a paper that is concurrently under review at ICLR2018, which I haven't read in depth. The paper compares against a Word2vec baseline that pools all the heterogeneous content learns just one set of embeddings. Results are shown on IMDB movies and a proprietary education platform datasets. In both the tasks, Feat2vec leads to significant reduction in error compared to Word2vec.

Comments:

The paper is well written and addresses an important problem of learning word embeddings when there is inherent structure in the feature space. It is a very practically relevant problem. The novelty of the proposed approach seems limited in light of the related paper that is concurrently under review at ICLR2018, on which this paper heavily relies. Perhaps the authors should consider combining the two papers into one complete paper? The structured deep-in factorization machines allow higher-level interactions in embedding learning which allows the authors to learn embeddings for heterogeneous set of features. The sampling approaches proposed seem pretty straightforward adaptations of existing methods and not novel enough.

---

> ### Author Response · Authors · 2018-01-05
> **Stronger contribution (better novelty) is in place**
>
> Thank you for the constructive comments. Your main criticism for the paper was that the contribution of our work was not significant enough to justify the two publications that we were aiming for. Following your suggestion, we have combined the two papers and added the relevant parts of the other paper (we only extended our submission with the results that would be relevant to the combined version).
>
> While the original paper only addressed unsupervised learning of embeddings,  the revised manuscript also addresses supervised learning of embeddings.  We demonstrate that our general supervised method can have better performance  than recently published single purpose methods (DeepCoNN and Collaborative Topic Regression) on two publicly available datasets, Yelp and CiteULike.  We also explain in more detail how Feat2Vec extends Factorization Machines.
>
> We hope that this major revision address your reservations.

---

### Official Review · AnonReviewer3 · 2017-11-27
**Not clear what I can learn from this**

**Rating:** 2
**Confidence:** 2

**Review:**

SUMMARY.

The paper presents an extension of word2vec for structured features.
The authors introduced a new compatibility function between features and, as in the skipgram approach, they propose a variation of negative sampling to deal with structured features.
The learned representation of features is tested on a recommendation-like task.


----------

OVERALL JUDGMENT
The paper is not clear and thus I am not sure what I can learn from it.
From what is written on the paper I have trouble to understand the definition of the model the authors propose and also an actual NLP task where the representation induced by the model can be useful.
For this reason, I would suggest the authors make clear with a more formal notation, and the use of examples, what the model is supposed to achieve.

----------

DETAILED COMMENTS
When the authors refer to word2vec is not clear if they are referring to skipgram or cbow algorithm, please make it clear.
Bottom of page one: "a positive example is 'semantic'", please, use another expression to describe observable examples, 'semantic' does not make sense in this context.
Levi and Goldberg (2014)  do not say anything about factorization machines, could the authors clarify this point?
Equation (4), what do i and j stand for? what does \beta represent? is it the embedding vector? How is this formula related to skipgram or cbow?
The introduction of structured deep-in factorization machine should be more clear with examples that give the intuition on the rationale of the model.
The experimental section is rather poor, first, the authors only compare themselves with word2ve (cbow), it is not clear what the reader should learn from the results the authors got.
Finally, the most striking flaw of this paper is the lack of references to previous works on word embeddings and feature representation, I would suggest the author check and compare themselves with previous work on this topic.

---

> ### Author Response · Authors · 2018-01-05
> **Major revision - significantly improved**
>
> Thank you for your helpful comments.  Because another reviewer suggested merging our two ICLR submissions, we underwent a major revision of the paper and now have two main contributions -- this is, we can calculate embeddings in a supervised setting (labels are available), and in an unsupervised setting (labels are not available).
>
> You stated two main criticisms to the paper:
> * References. You mentioned that the most striking flaw of the paper is lack of references. We added roughly three times more citations (we increased references from ~12 to ~36). We believe that the paper is now much better situated in the literature.
> * Evaluation. To our knowledge we are the first ones to propose learning unsupervised embeddings for multiple feature types.  The Word2Vec algorithms are  other unsupervised  embedding methods (though, W2V only works with words), and that is why we compare with them.
> Because of the major revision of the paper, we believe we improved the empirical result section significantly. We added 2 additional datasets (total of 4), and added 4 baselines altogether (CBOW W2V, Matrix Factorization, Collaborative Topic Regression and  DeepCoNN)
>
>
> Other detailed comments:
> We removed the reference of Levy & Goldberg (but the general point is that factorization machines are a general case of matrix factorization)
> We rewrote the introduction to make more salient our contributions, and we believe that it is now more clear what the model achieves. We streamlined the notation. Additionally, we clarified the language surrounding Word2Vec.
>
> We hope that these major revisions address your reservations.

---

### Official Review · AnonReviewer1 · 2017-11-30
**Neat representation learning scheme for structured features**

**Rating:** 7
**Confidence:** 5

**Review:**

This paper provides a clean way of learning embeddings for structured features that can be discrete -- indicating presence / absence of a certain quality. Further, these features can be structured i.e. a set of them are of the same 'type'. Unlike, word2vec there is no hard constraint that similar objects must have similar representations and so, the learnt embeddings reflect the likelihood of the observed features. Therefore, this can be used as a multi-label classifier by using two feature types -- the input and the set of categories. This proposed scheme is evaluated on two datasets -- movies and education in a retrieval setting.

I would like to see an evaluation of these features in a classification setting to further demonstrate the utility of these embeddings as compared to directly embedding the discrete features and then performing a K-way classification. For example, I am aware of -- http://manikvarma.org/downloads/XC/XMLRepository.html contains some interesting datasets which have a large number of discrete features and classes.

---

> ### Author Response · Authors · 2018-01-05
> **Thank you**
>
> Thank you for your informative comments on our paper. We have added experiments for supervised Feat2Vec, which includes a multi-label prediction task on a public dataset (CiteULike) benchmarked against other state of the art methods. We hope that this experiment at least partially addresses your desire to see Feat2Vec in a K-way classification task.  We would also like to point you to the ranking task done classifying the director of a film based on its task members. The 2.43% Top-1 Precision can be imagined as the performance of the unsupervised F2V embedding algorithm on a K-way classification task (as compared to Word2Vec’s CBOW algorithm).

---

### Author Response · Authors · 2018-01-05
**Summary for major revision**

Dear Chair,

The two main criticisms of the paper by the reviewers were (i) lack of references and (ii) that the results of the study were not significant enough to justify the two publications that we were aiming for.  For this reason, we added roughly 3x more citations to the paper to better situate the contributions in the literature.  Additionally, we merged this submission with our other concurrent ICLR manuscript.

We are hoping that we can get an opportunity to share our results with the ICLR community.  In the unsupervised setting, our work  is the first one to enable leveraging arbitrary feature types (a more general approach than exists in the literature).   In the supervised scenario, we provide evidence that our general method can have better performance than ad-hoc networks that work for a single purpose.

Thanks,
Authors

---

### Public Comment · (anonymous) · 2018-01-24
**Missing Related Work**

The paper makes claims about being the "first algorithm that is able to calculate general-purpose embeddings that are not tuned for a single specific prediction task for arbitrary features". I don't think this is true:

This paper "Exponential family embeddings" (https://arxiv.org/pdf/1608.00778.pdf) was in NIPS 2016 and presents a principled approach to handling various feature types.

-Leveraging structure/groups in the data. I think this was popular a few years ago in the matrix factorization / dictionary learning / sparse learning community e.g. the references in:
https://www.di.ens.fr/~fbach/talk_sparse_pca_DL_online.pdf

but the authors don't seem to mention any of this work.

Thus, I find the novelty of the paper limited.


(2) Evaluation. I am not persuaded that some of the experiments are standard evaluation, particularly Section 4.2 "General purpose embeddings".  For instance they take a movie dataset (IMDB) and compare the similarity of movie directors to those of actors who were cast in the same film. I don't think that is standard.

Perhaps the authors could consider some of the data/tasks used in the following papers to evaluate the nature of their embeddings compared to word2vec.

1. https://arxiv.org/abs/1411.4166

2. https://nlp.stanford.edu/pubs/glove.pdf

3. https://arxiv.org/pdf/1608.00778.pdf

---

> ### Author Response · Authors · 2018-01-24
> **Key distinctions between our work and prior work**
>
> Thank you for your insightful comments.
>
> I. NOVELTY
> After reviewing your two references, we believe that our novelty claims still stand:
>
> 1) Regarding the "exponential family embeddings," our claim refers to general-purpose embeddings, which we define as “embeddings of an unsupervised method that can be used for a variety of auxiliary prediction tasks.” Therefore, our novelty claim is about unsupervised learning of embedding models with features, while the paper that you link to is a supervised approach.
>
> 2) The "structured factorization" work that you point out is a way to introduce structured sparsity, and could be used in tandem with our method. We define the structure in the loss function (not as regularization), as a novel way to combine features to get embeddings at different levels of granularity.
>
> While structured PCA requires groups of features to disappear simultaneously, the features that remain in the model are jointly projected to a common space. The "embeddings" that structured PCA discovers are a mixture of the remaining features. On the other hand, our approach can find an embedding for *each* value of the different *group* of features. Thus, PCA finds different vectors that are useful for a fundamentally different problem.
>
> We were unaware of these works and agree they should be cited in a published version of our work. We will include these references in a future revision.
>
> II. EVALUATION
> Our evaluation for Unsupervised Feat2Vec differs from the standard evaluations for word embeddings since we are not dealing with language data - for example, word analogy is not applicable for the IMDB dataset. The evaluations used for supervised methods such as exponential family embeddings are also not applicable, as in our case there is no specific prediction task the embeddings are tuned for. Since building unsupervised embeddings for arbitrary feature types is not a well-studied problem, we are unaware of any standard way to evaluate them.
>
> Thanks again for reading our work. I hope this response addresses your reservations.

---

### Decision · Program_Chairs · 2018-01-29
**ICLR 2018 Conference Acceptance Decision**

**Decision:**

Reject

**Comment:**

The paper presents an approach for learning continuous-valued vector representations combining multiple input feature sets of different types, in both unsupervised and supervised settings.  The revised paper is a merger of the original submission and another ICLR submission.  This meta-review takes into account all of the comments on both submissions and revisions.

The merged paper is an improvement over the two separate ones.  However, the contribution over previous work is still a bit unclear.  It still does not sufficiently compare to/discuss in the context of other recent work on combining multiple feature groups.

The experiments are also quite limited.  The idea is introduced as extremely general, but the experiments focus on a small number of specific tasks, some of them non-standard.